# Transfection of hPSC-Cardiomyocytes Using Viafect™ Transfection Reagent

**DOI:** 10.3390/mps3030057

**Published:** 2020-08-09

**Authors:** Sara E. Bodbin, Chris Denning, Diogo Mosqueira

**Affiliations:** Division of Cancer & Stem Cells, Biodiscovery Institute, University of Nottingham, Nottingham NG7 2RD, UK

**Keywords:** human pluripotent stem cell-derived cardiomyocytes, transfection, Viafect^TM^, disease modelling

## Abstract

Twenty years since their first derivation, human pluripotent stem cell-derived cardiomyocytes (hPSC-CMs) have shown promise in disease modelling research, while their potential for cardiac repair is being investigated. However, low transfection efficiency is a barrier to wider realisation of the potential this model system has to offer. We endeavoured to produce a protocol for improved transfection of hPSC-CMs using the Viafect^TM^ reagent by Promega. Through optimisation of four essential parameters: (i) serum supplementation, (ii) time between replating and transfection, (iii) reagent to DNA ratio and (iv) cell density, we were able to successfully transfect hPSC-CMs to ~95% efficiencies. Transfected hPSC-CMs retained high purity and structural integrity despite a mild reduction in viability, and preserved compatibility with phenotyping assays of hypertrophy. This protocol greatly adds value to the field by overcoming limited transfection efficiencies of hPSC-CMs in a simple and quick approach that ensures sustained expression of transfected genes for at least 14 days, opening new opportunities in mechanistic discovery for cardiac-related diseases.

## 1. Introduction

Transfection is the introduction of foreign DNA into a cell to assess gene expression, evaluate gene silencing and for the analysis of recombinant proteins [1]. Transfected cells enable the study of genes and their functions by either repression or overexpression and to produce recombinant proteins. The first transfection protocols date back to the 1960’s by Pagano and Vaheri [2]. Since then, numerous methods have been developed which are organised into three groups: chemical, physical and biological [3]. The introduction of DNA is either transient or stable, resulting in: a shorter expression window due to non-integration of the material to the genome vs. continuous expression through the passing of DNA onto the progeny [4]. The objective of the experiment (or the cell type) determines the choice between the two approaches. Transient methods can be affected by loss of transfected DNA through cell division or by degradation by nucleases, whereas maintaining stable expression can be hindered by toxicity of the transgene or silencing by gradual epigenetic mechanisms, particularly when regulatory pathways are involved [5,6,7].

Efficient transfection methods can greatly support disease modelling research to clarify genetic causation and disrupt pathological signalling pathways. Despite accounting for 31% of global deaths worldwide in 2017 [8], the disease mechanisms of many cardiac disorders are poorly understood, especially those in the cardiomyopathies [9]. At present, human pluripotent stem cell (hPSC) derived cardiomyocytes (CM) might be one of the best resources to promote cardiac repair [10,11] and model cardiac disorders, complementing preceding in vivo models [9,12,13]. Transfection of hPSC-CMs provides further mechanistic insight into these diseases, towards the development of improved and effective treatments, which are still elusive in several cardiac disorders such as cardiomyopathies [14]. For instance, the overexpression of long non-coding RNA (lncRNA) *Ahit* in mice was performed to modulate the hypertrophic response and to identify interacting partners and novel disease mechanisms [15]. Additionally, sustained overexpression of genetically-encoded calcium or voltage sensors may be used to establish reporter cell lines of hPSC-CMs, which currently require suboptimal genome integration or short-lived dyes [16,17]. Finally, overexpression may also be used to promote more efficient cardiomyocyte differentiation and maturation [18]. Altogether, hPSC-CM transfection can greatly benefit the research community, particularly if high efficiency protocols are available.

Nevertheless, there is an unmet need to improve transfection protocols in hPSC-CMs, which presents a barrier to full realization of the opportunities for these cells as model systems and research tools into cardiac disorders. This is a critical requirement, as a PubMed search reflects an exponential increase in publications on the subject of human induced pluripotent stem cells (hiPSC) and also hiPSC-CMs, but displays a severe deficiency in the number of papers on the transfection of hiPSC-CMs (Figure 1). In comparison to the 2133 publications on the topic of hiPSC-CMs since 2000, there are only 85 publications referring to transfection with even best-in-class physical or chemical methods achieving between 20% and 56%, respectively [19,20]. The importance of obtaining a high transfection efficiency (TE) for enhanced mechanistic understandings of disease compels us to explore better transfection methods for hPSC-CMs.

Determining which transfection protocol to employ requires the consideration of several factors: (i) proliferation rate of cell type of interest, (ii) cost-effectiveness, (iii) compatibility with downstream experiments and (iv) the desirable transfection efficiency (TE) level. Most kits and reagents are expensive and will require some degree of optimisation. Physical techniques (such as electroporation or nucleofection) can cause cell death through disruption of the cell membrane and might also require the purchase of specialist equipment [22]. Some chemical procedures such as lipofection, the use of liposomes to fuse to cell membranes of the target cells, can affect expression of the introduced DNA, cause chemical toxicity to some cell types and also reduce the TE [23]. Finally, biological methods such as viral vectors can be exceptionally hazardous to the operator and can also have mutagenic effects in the targeted cells due to unintended insertional mutagenesis [24,25]. Moreover, some cell lines are more challenging to transfect [3]. Certain biological properties of cells can make the uptake of DNA difficult, e.g., those that grow in clumps which display reduced cell membrane area availability, as well as highly confluent cells. Another desirable feature for effective stable transfection is constant nuclear envelope reformation due to rapid division, which hPSC-CMs do not possess since they are largely quiescent cells [26].

Through systematic optimization, we have found that Viafect™ is a reagent that can provide very high efficiency for the transfection of cardiomyocytes. This cationic delivery reagent in an aqueous solution functions by interacting with the negatively charged DNA, allowing its passage into the cell by preventing the electrostatic repulsion of the cell membrane [27]. The transfection of CMs using Viafect™ is transient as the DNA inserted does not integrate into the genome. However, hiPSC-CMs show very limited proliferation rates, thus preventing the dilution of the transfected DNA over time [28]. Our overarching goal of transfecting hPSC-CMs is to identify novel targets that may be underlying molecular mechanisms of cardiomyopathy progression.

Herein, we describe the steps for high TE of hPSC-CMs, with a focus on optimisation of serum content, reagent:DNA ratio, and the importance of seeding density and timing. Optimisation was carried out in CMs derived from an in-house cell line known as REBL-PAT, an hiPSC line previously described [29]. CMs derived from a human Embryonic Stem Cell (hESC) line known as HUES7 [30] were also used to validate this method in independent CM lines. While the determined key parameters (serum supplementation, time between replating and transfection, reagent:DNA ratio and cell density) are generally applicable across various hPSC lines, the end user should optimise them due to the existence of various hPSC-CM culture protocols [31].

## 2. Materials and Methods

### 2.1. Materials

Vitronectin–Recombinant Human Protein, 500 µg/mL in PBS, (Gibco™, Thermofisher, Loughborough, UK, Cat. No: A174700)Phosphate Buffer Saline (PBS, Gibco™, Thermofisher, Loughborough, UK, Cat. No: 14190–094)Nunc 96-well Flat Bottom, Delta Coated (Thermofisher, Loughborough, UK, Cat. No: 167008)B-27™ Supplement (50X), custom (Gibco™, Thermofisher, Loughborough, UK, Cat. No: 0080085SA)RPMI 1640 medium (Gibco™, Thermofisher, Loughborough, UK, Cat. No: 21875034)Fetal Bovine Serum (FBS, Gibco™, Thermofisher, Loughborough, UK, Cat. No: 11573397)Viafect™ (Promega, Southampton, UK, Cat. No: E4981 1X 0.75 mL, E4982 2X 0.75 mL)Opti-MEM™ (Gibco™, Thermofisher, Loughborough, UK, Cat. No: 31985070)Axygen™ MaxyClear Snaplock Eppendorff tubes, 1.5 mL (Axygen™, FisherScientific, Loughborough, UK, Cat. No: 11326144)Concentrated plasmid DNA of interest (100 ng per well of a 96 well plate is required); if using more than one plasmid divide by the number of plasmids to ensure total of 100 ng per wellpmaxGFP from Lonza™ P3 Primary Cell 4D-Nucleofector™ X Kit L, (FisherScientific, Loughborough, UK, Cat. No: 11326144), or any other eGFP reporter plasmid high content imagingCellavista^®^ (Synentec, BPES, Kent, UK) or any other fluorescence plate imaging system

### 2.2. Methods

Details on culture, differentiation and dissociation of hPSC-CMs can be found in the Supplementary Methods. All the steps were performed in sterile conditions in a type II Biological Safety Cabinet, and cells were maintained in a humidified incubator, at 37 °C and 5% CO_2_ (Heracell).

#### 2.2.1. Replating hPSC-CMs

Prepare a solution of Vitronectin at 5 µg/mL in PBS (1:100 dilution from stock). A total of 5 mLs is enough to coat a whole 96 well plate.Coat a Nunc 96-Well Flat Bottom Delta plate with 50 µL of Vitronectin solution at 5µg/mL. Incubate at room temperature (RT) for 1 h.Supplement RPMI 1640 with a B27 supplement (RB27, 1:50 dilution from the stock), aliquot enough for 75 µL/well X the number of wells to be used and let it reach RT.**OPTIONAL STEP** you may use other culture media and extracellular matrix proteins provided they have been optimised for cardiomyocyte culture.Aspirate the Vitronectin and wash with 50 μL PBS. Aspirate PBS and add 75 µL RB27 per well.Resuspend pelleted pre-dissociated hPSC-CMs at 1 × 10^6^ cells/mL in RB27, forming a single cell suspension.Replate hPSC-CMs at a density ranging from 3.0 × 10^4^ to 4.0 × 10^4^ per well (~95–125 K/cm^2^) in wells containing 75 μL RB27. Allow the cells to settle for 30 min at RT before placing in the incubator.Replace medium with 100 μL/well RB27 every other day for 3 days post plating.

#### 2.2.2. Transfecting hPSC-CMs

On the morning of transfection, replace media with 90 μL/well RB27 supplemented with 10% FBS.Calculate transfection mixture so that each well of a 96 well plate is transfected with 100 ng of plasmid DNA: Volume of plasmid(s) = $\frac{100\;\mathrm{ng}/\mathrm{well}\times \mathrm{number}\;\mathrm{of}\;\mathrm{wells}}{\mathrm{Plasmid}\;\mathrm{concentration}\;\left(\mathrm{ng}/\mu l\right)}$Volume of Viafect (μL) (6:1 ratio) = 6×total plasmid mass in μgVolume of OptiMEM (μL) = $\left(\mathrm{number}\;\mathrm{wells}\;\times 10\right)-V_{\mathrm{plasmid}}-{\;V}_{\mathrm{Viafect}}$Before pipetting, allow the reagents to reach RT and mix by inverting.On the afternoon of transfection, add the reagent volumes calculated in step 9 to a 1.5 mL Eppendorff tube.
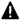
**CRITICAL STEP** Flick tube to homogenise transfection mixture (do not vortex or pipette mix). Incubate at RT for 20 min.Add 10 μL of transfection mixture to each well, directly into the medium without touching the cells.Place cells in an incubator at 37 °C and 5% CO_2._ Replace media the following day with RB27 with or without FBS (as per your standard culture practises) and then every second day from thereon until performing desired phenotypic assay.Measure transfection efficiency using an appropriate reporter gene (e.g., GFP), preferably using a high content imaging system.

## 3. Results

The transfection of hPSC-CMs using Viafect™ was optimised in four main parameters: (i) serum content, (ii) the day of transfection after replating, (iii) ratio of Viafect™ to DNA and (iv) the cell seeding density (Table 1).

hPSC-CMs were generated following previously published methods achieving >95% purity [29], quantified by high-content imaging of α-actinin (Appendix A), as detailed in [32]. Cardiomyocytes were then transfected with a pmaxGFP reporter plasmid and the transfection efficiency was evaluated by measuring the percentage of cells expressing GFP using fluorescent imaging (Figure 2a). Optimisation was carried out in two steps. First, different ratios of Viafect™:DNA were compared, versus the amount of FBS in the media (Figure 2b). The 6:1 Viafect to DNA ratio in hiPSC-CMs cultured in RB27 supplemented with 10% FBS yielded the greatest TE (~95%, Figure 2c), whilst maximizing cost-effectivity. Subsequently, we assessed how the day of transfection after replating affected TE. We observed that 3 days post-dissociation maximises transfection efficiency by allowing cardiomyocytes enough time to attach and recover from the dissociation procedure and be receptive to uptake DNA. Following our optimisation, we assessed how long expression could be sustained for in hiPSC-CMs, considering the transient transfection nature of this transfection. For this, cells were transfected with a strong promoter plasmid driving the expression of a lncRNA being investigated for its potential role in hypertrophic cardiomyopathy. The fold change of expression was quantified by qPCR at 4, 7 and 14 days post transfection. Surprisingly, we observed that lncRNA overexpression was maintained in hiPSC-CMs for up to at least 14 days (Figure 2d).

2D hPSC-CM phenotyping assays rely mostly on cell clusters or monolayers [32,33]. Thus, different hPSC-CM densities were tested to ensure compatibility with those assays. We observed that optimum density levels ensuring high TE (above 90%) were in the range of 30–40 K hiPSC-CMs per well (96 well format, Figure 3), which translates to an approximately 80% confluence, whilst also avoiding drawbacks of high-density cultures (50 K/well) where cells tend to peel off and detach. We settled on an average of 35,000 cells per well of a 96 well plate, as it provides an adequate pool for further experiments. However, while the transfection of hiPSC-CMs did not change the high purity (Appendix A), the total number of cardiomyocytes decreased by ~34%, indicating ~66% cell viability (Appendix A).

Furthermore, we confirmed that the conditions identified above were readily transferrable to cardiomyocytes derived from other human pluripotent stem cells by demonstrating high TE in the HUES7-CMs, an hESC line (Figure 4) [30].

## 4. Discussion

We optimised a quick and simple method for attaining high transfection efficiencies (~95%) in hiPSC-and hESC-derived cardiomyocytes. In summary, the optimal conditions were: medium supplementation with 10% FBS, a ratio of Viafect™:DNA at 6:1 and a seeding density ranging between 30,000 and 40,000 cells/well in a 96 well plate. We observed that the time to transfect the cardiomyocytes achieving highest efficiency was three days post dissociation, and that this was a critical factor in ensuring a high TE. Longer timelines between replating and transfection were detrimental, just as performing transfection too soon after replating. It is worth noting that 10% FBS at Viafect™ to DNA ratios 6:1 and 8:1 was virtually identical. We opted for 6:1 to reduce usage of Viafect™ for economical purposes. This protocol is suitable for both protein-coding genes (as we have optimized it using a GFP reporter plasmid) and non-coding DNA.

While high efficiency transfection (>80%) of undifferentiated hiPSCs was achieved long ago [34], the same is not true of hiPSC-CMs, which have proved refractory to simple approaches for gene transfer. For example, other chemical and physical methods using Lipofectamine and magnetic nanoparticles achieved a maximum TE of up to ~56% and ~20%, respectively, in hiPSC-CMs [19,20]. Viral gene transfer methods have shown more success, with adeno associated virus (AAV), adenovirus and lentivirus achieving 90–95% in hiPSC-CMs [35], but often causing cell toxicity, and entailing technical complexity of viral engineering and/or biosafety risks. Therefore, finding a simple chemical approach for transfection of hiPSC-CMs provides a benefit to the research community. Viafect^TM^ could achieve ~95% TE in a rapid and simple protocol, overcoming suboptimal efficiencies of other physical and chemical methods, without the need for lengthy pre-complexation procedures and/or magnetic activation necessitating specialised equipment. Nevertheless, Viafect transfection resulted in ~66% cardiomyocyte viability. While a previous method using liposomes [20] achieved higher cardiomyocyte viabilities upon transfection (80.8–92.8%), the transfection efficiency reported was much lower (~56%) than when using Viafect (~95%). Importantly, the cell loss upon transfection did not prevent further phenotypic assays, such as the identification of hypertrophy markers (such as Brain Natriuretic Peptide, BNP) (Appendix A).

Ordinarily, transiently transfected cells are collected 24–72 h after transfection as the expression of the introduced gene is short lived due to factors such as cell division, half-life of gene expression and protein turnover [36]. Extensive research has been conducted to determine whether CMs truly are non-dividing. The previously accepted assumption that there are a fixed number of cardiomyocytes in a human heart has been questioned, with more recent data showing that less than 1% of cardiomyocytes are replaced per year in an adult heart [37], reaffirming that CMs possess little ability to divide. This has been corroborated by data with hiPSC-CM lines [38]. The limited cell division is what we consider to have been key to our ability to maintain expression of any gene of interest for up to at least two weeks in our hiPSC-CMs transfected with Viafect™, which was not shown in other non-viral methods. With no active mitotic events that may end up diluting the foreign DNA, the only factor limiting fully stable transfection is the degradation of the plasmid DNA over time. Thus, this method enables highly efficient sustained expression of the plasmid of interest without integration into the genome and the shortcomings associated with the use of lentiviral vectors [39].

Our standard culture practices avoid the use of serum due the masking effects it may have on hypertrophy [40], which would interfere with our experiments. Despite this, we determined that the use of serum for a short exposure time improves the TE, possibly by improving the uptake of DNA. The serum is removed at the following medium change (24 h post transfection) and we have not observed any detrimental effects such as cytotoxicity or clear changes in cell morphology upon serum withdrawal. As such, the use of FBS can be restricted for the duration of the transfection, avoiding longer term effects. The majority of cationic delivery methods do not work in cultures with serum present due to the effects on DNA–lipid complex formation which is essential for transfection to take place and thus reduce the TE [41]. For cells that must be grown with serum, Viafect™ provides an excellent alternative.

In conclusion, we determined that Viafect™ can successfully transfect hiPSC-CMs with high transfection efficiencies with no apparent undesired effects or clear interference with our experimental outcomes. We confirmed the reagent is also useful to successfully transfect hESC-CMs. We also observed that the lack of CM division is favourable in order to carry out longer term experiments in CMs, which are inherently restricted in transient transfection methods. Altogether, our protocol is novel because it overcomes low efficiencies of other hPSC-CM transfection methods, retaining high cardiomyocyte purity and compatibility with downstream phenotypic assays. Moreover, despite a mild reduction in cell viability, this simple and cost-effective method enables sustained expression of the transfected gene for at least 14 days. Ultimately, this protocol is expected to greatly support the field of cardiac disease modelling or any other applications where hPSC-CMs play a pivotal role (e.g., transplantation and drug discovery) [42].

## Figures and Tables

**Figure 1 mps-03-00057-f001:**
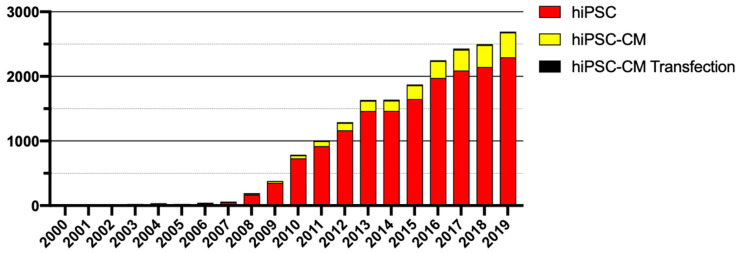
Publications by year in regard to human pluripotent stem cells (hiPSC)s, human pluripotent stem cell-derived cardiomyocytes (hiPSC-CM)s and their transfection [21].

**Figure 2 mps-03-00057-f002:**
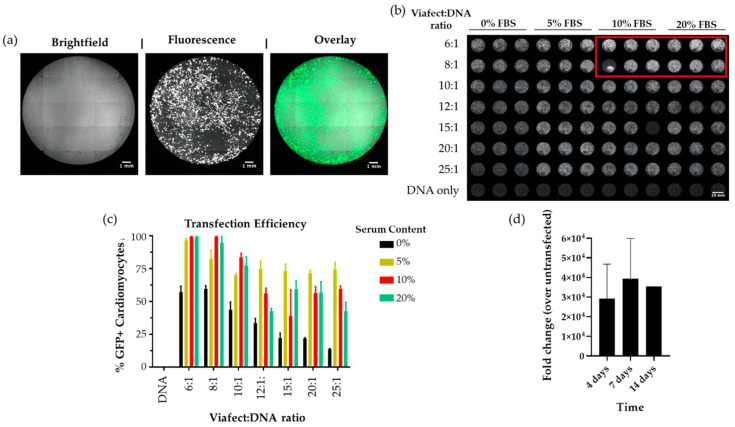
Optimisation of hiPSC-CM transfection with Viafect™. To determine the experimental conditions ensuring highest TE, hiPSC-CMs were replated in 96 well plate format and transfected with reporter plasmid pmaxGFP under several combinations of serum content and Viafect:DNA ratios. (**a**) High content fluorescence imaging enabled determination of transfection efficiency (TE) in hiPSC-CMs; (**b**) Optimisation of serum content and Viafect to DNA ratio in 96 well plate format revealed that (**c**) transfection with 6:1 Viafect:DNA ratio in medium containing 10% foetal bovine serum (FBS) achieves >95% TE; (**d**) Sustained expression of a lncRNA in hiPSC-cardiomyocytes. The results were normalised to an untransfected control. *n* = 3 biological replicates (4 days and 7 days), *n* = 1 (14 days). Scale bars in (a) = 1 mm, (b) = 10 mm.

**Figure 3 mps-03-00057-f003:**
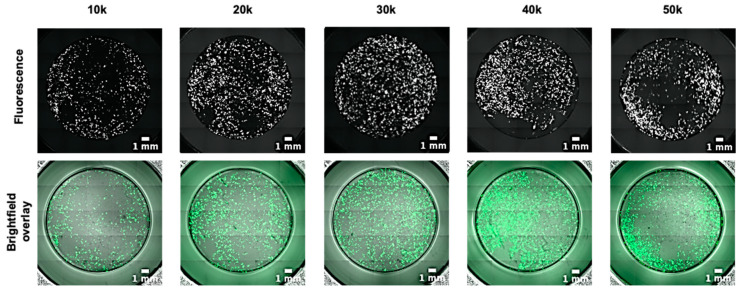
Seeding density affects transfection efficiency. hiPSC-CMs were seeded at a range of densities, from 1 × 10^4^ to 5 × 10^4^ per well, then transfected with 6:1 Viafect to DNA with 10% serum supplementation, 3 days post replating. Live images taken 3 days post transfection (upper panel GFP only, lower panel GFP and brightfield overlay). Scale bar = 1 mm.

**Figure 4 mps-03-00057-f004:**
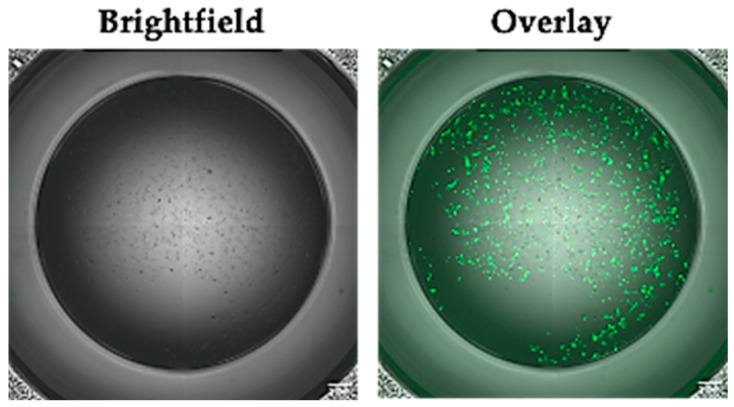
hESC-CMs can be successfully transfected with Viafect. Live images showing brightfield and GFP fluorescence. hESC-CMs were transfected with the reporter plasmid pmaxGFP, using the same parameters of serum, content, reagent: DNA ratio, seeding density and timing determined by the hiPSC line. Scale bar = 1 mm.

**Table 1 mps-03-00057-t001:** Optimal conditions for transfecting hPSC-CMs with high efficiency.

Condition	Optimal	Range Tested
Serum content	10% FBS	0−20%
Day of Transfection after replating	3 days	1–7 days
Ratio of Viafect™ (μL) to DNA (mass)	6:1	6:1-25:1
Seeding Density per well (96 well plate)	30–40 K	10–50 K

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
