# Peer review of "Transfection of hPSC-Cardiomyocytes Using Viafect™ Transfection Reagent"

_mps, 2020, doi:10.3390/mps3030057_

Round 1

Reviewer 1 Report

Bodbin et al, study “Transfection of hiPSC-cardiomyocytes using 2 Viafect™ Transfection Reagent” and their analysis revealed ~95% transfection efficiencies but in the manuscript no convincing data provided in results in the present study. This study is suggesting that after review PubMed literature they found there are 85 publications referring to transfection between 20% and 56%, respectively by Wang et al, 2018 and Tan et al, 2019. This ~95% transfection efficiency achieved for presentation a lesser amount of data to coming conclusion or other publications provide more data to show less efficiency need further addressed by authors in this paper.   Authors appealing the transfection of CMs using Viafect™ is transient as the DNA inserted does not integrate into the genome. However, hiPSC-CMs show very limited proliferation rates, thus preventing the dilution of the transfected DNA over time Zhang et al, 2009. In the last 11 years of research, there are a few modifications to these protocols to achieve 99% pure cardiomyocytes using various methods and each method has its merits vs demerits, in the present paper completely ignored their purity of cardiomyocytes or no data provide in the manuscript. They need to provide flow cytometry data of cardiomyocyte purity TnT before and after transfection.

But authors showing how long expression of HCM marker they called (lncRNA : need more details of lncRNA type and their transfection protocol) to transient transfection nature of this transfection. lncRNA which do not overlap protein-coding genes it’s may not suitable marker for transfection studies if this reagent wants to commercialize for protein-coding genes, need to choose another marker.   The authors could further enhance the manuscript by addressing the following points. If they postfix and perform immunocytochemistry to show % CM comparison compared to GFP (green) TnT (red) and NKX2.5 immunofluorescent analysis will further support their conclusions. If they provide a high magnification picture of their transfected cardiomyocyte shape will help the reader will understand this is cardiomyocyte transfection study, as their low-density cardiomyocyte has less effect of cell cycle and cell survival.

In the results, part authors need to provide data about their conclusions as they confirmed the reagent is also useful to successfully transfect hESC-CMs.  They mentioned we also observed that the lack of CM division is favorable in order to carry out longer-term experiments in CMs, which are inherently restricted in transient transfection methods, more details need to give to understand the importance of this study to UK public and funding agencies.

In the results, part authors need to provide data about their conclusions cell viability and their analysis submit as supplemental raw data as supplement files.

Reviewer 2 Report

The manuscript titled " Transfection of ....Transfection Reagent" authored by Bodbin et al. shows that Viafect from Promega is a suitable transfecting reagent, especially for the difficult-to-transfect hiPSC-cardiomyocytes. The authors very thoroughly optimized parameters and successfully transfected these cells at ~95% efficiency. This method will help the field to move forward.

This reviewer has no concerns and recommends that this manuscript be accepted for publication.

Reviewer 3 Report

It is not clear from the research exactly where your transvection method is better. The methods can be spelled out in more detail. And of course, how exactly hiPSC-CMs transvection will help provides a benefit to the research community.

Round 2

Reviewer 1 Report

The authors describe New or Novelty in this study
